# Transfer Learning from Speaker Verification to Multispeaker Text-To-Speech Synthesis

**Ye Jia**[*]   **Yu Zhang**[*]   **Ron J. Weiss**[*]   **Quan Wang**   **Jonathan Shen**   **Fei Ren**
**Zhifeng Chen**   **Patrick Nguyen**   **Ruoming Pang**   **Ignacio Lopez Moreno**   **Yonghui Wu**
Google Inc.
{jiaye,ngyuzh,ronw}@google.com

## Abstract

We describe a neural network-based system for text-to-speech (TTS) synthesis that is able to generate speech audio in the voice of different speakers, including those unseen during training. Our system consists of three independently trained components: (1) a *speaker encoder network*, trained on a speaker verification task using an independent dataset of noisy speech without transcripts from thousands of speakers, to generate a fixed-dimensional embedding vector from only seconds of reference speech from a target speaker; (2) a sequence-to-sequence *synthesis network* based on Tacotron 2 that generates a mel spectrogram from text, conditioned on the speaker embedding; (3) an auto-regressive WaveNet-based *vocoder network* that converts the mel spectrogram into time domain waveform samples. We demonstrate that the proposed model is able to transfer the knowledge of speaker variability learned by the discriminatively-trained speaker encoder to the multispeaker TTS task, and is able to synthesize natural speech from speakers unseen during training. We quantify the importance of training the speaker encoder on a large and diverse speaker set in order to obtain the best generalization performance. Finally, we show that randomly sampled speaker embeddings can be used to synthesize speech in the voice of novel speakers dissimilar from those used in training, indicating that the model has learned a high quality speaker representation.

## 1   Introduction

The goal of this work is to build a TTS system which can generate natural speech for a variety of speakers in a data efficient manner. We specifically address a zero-shot learning setting, where a few seconds of untranscribed reference audio from a target speaker is used to synthesize new speech in that speaker's voice, without updating any model parameters. Such systems have accessibility applications, such as restoring the ability to communicate naturally to users who have lost their voice and are therefore unable to provide many new training examples. They could also enable new applications, such as transferring a voice across languages for more natural speech-to-speech translation, or generating realistic speech from text in low resource settings. However, it is also important to note the potential for misuse of this technology, for example impersonating someone's voice without their consent. In order to address safety concerns consistent with principles such as [1], we verify that voices generated by the proposed model can easily be distinguished from real voices.

Synthesizing natural speech requires training on a large number of high quality speech-transcript pairs, and supporting many speakers usually uses tens of minutes of training data per speaker [8]. Recording a large amount of high quality data for many speakers is impractical. Our approach is to decouple speaker modeling from speech synthesis by independently training a speaker-discriminative embedding network that captures the space of speaker characteristics and training a high quality TTS

---

[*]Equal contribution.

model on a smaller dataset conditioned on the representation learned by the first network. Decoupling the networks enables them to be trained on independent data, which reduces the need to obtain high quality multispeaker training data. We train the speaker embedding network on a speaker verification task to determine if two different utterances were spoken by the same speaker. In contrast to the subsequent TTS model, this network is trained on untranscribed speech containing reverberation and background noise from a large number of speakers.

We demonstrate that the speaker encoder and synthesis networks can be trained on unbalanced and disjoint sets of speakers and still generalize well. We train the synthesis network on 1.2K speakers and show that training the encoder on a much larger set of 18K speakers improves adaptation quality, and further enables synthesis of completely novel speakers by sampling from the embedding prior.

There has been significant interest in end-to-end training of TTS models, which are trained directly from text-audio pairs, without depending on hand crafted intermediate representations [17, 23]. Tacotron 2 [15] used WaveNet [19] as a vocoder to invert spectrograms generated by an encoder-decoder architecture with attention [3], obtaining naturalness approaching that of human speech by combining Tacotron's [23] prosody with WaveNet's audio quality. It only supported a single speaker.

Gibiansky et al. [8] introduced a multispeaker variation of Tacotron which learned low-dimensional speaker embedding for each training speaker. Deep Voice 3 [13] proposed a fully convolutional encoder-decoder architecture which scaled up to support over 2,400 speakers from LibriSpeech [12].

These systems learn a fixed set of speaker embeddings and therefore only support synthesis of voices seen during training. In contrast, VoiceLoop [18] proposed a novel architecture based on a fixed size memory buffer which can generate speech from voices unseen during training. Obtaining good results required tens of minutes of enrollment speech and transcripts for a new speaker.

Recent extensions have enabled few-shot speaker adaptation where only a few seconds of speech per speaker (without transcripts) can be used to generate new speech in that speaker's voice. [2] extends Deep Voice 3, comparing a *speaker adaptation* method similar to [18] where the model parameters (including speaker embedding) are fine-tuned on a small amount of adaptation data to a *speaker encoding* method which uses a neural network to predict speaker embedding directly from a spectrogram. The latter approach is significantly more data efficient, obtaining higher naturalness using small amounts of adaptation data, in as few as one or two utterances. It is also significantly more computationally efficient since it does not require hundreds of backpropagation iterations.

Nachmani et al. [10] similarly extended VoiceLoop to utilize a target speaker encoding network to predict a speaker embedding. This network is trained jointly with the synthesis network using a contrastive triplet loss to ensure that embeddings predicted from utterances by the same speaker are closer than embeddings computed from different speakers. In addition, a cycle-consistency loss is used to ensure that the synthesized speech encodes to a similar embedding as the adaptation utterance.

A similar spectrogram encoder network, trained without a triplet loss, was shown to work for transferring target prosody to synthesized speech [16]. In this paper we demonstrate that training a similar encoder to discriminate between speakers leads to reliable transfer of speaker characteristics. Our work is most similar to the speaker encoding models in [2, 10], except that we utilize a network independently-trained for a speaker verification task on a large dataset of untranscribed audio from tens of thousands of speakers, using a state-of-the-art generalized end-to-end loss [22]. [10] incorporated a similar speaker-discriminative representation into their model, however all components were trained jointly. In contrast, we explore transfer learning from a pre-trained speaker verification model.

Doddipatla et al. [7] used a similar transfer learning configuration where a speaker embedding computed from a pre-trained speaker classifier was used to condition a TTS system. In this paper we utilize an end-to-end synthesis network which does not rely on intermediate linguistic features, and a substantially different speaker embedding network which is not limited to a closed set of speakers. Furthermore, we analyze how quality varies with the number of speakers in the training set, and find that zero-shot transfer requires training on thousands of speakers, many more than were used in [7].

## 2 Multispeaker speech synthesis model

Our system is composed of three independently trained neural networks, illustrated in Figure 1: (1) a recurrent *speaker encoder*, based on [22], which computes a fixed dimensional vector from a speech

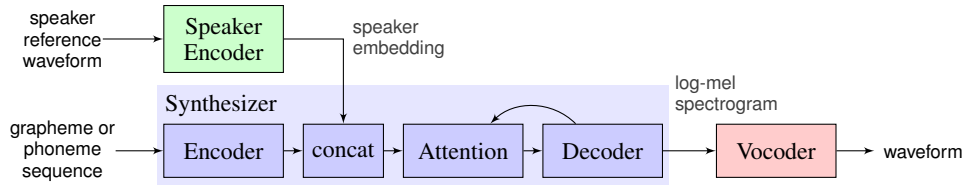

Figure 1: Model overview. Each of the three components are trained independently.

signal, (2) a sequence-to-sequence *synthesizer*, based on [15], which predicts a mel spectrogram from a sequence of grapheme or phoneme inputs, conditioned on the speaker embedding vector, and (3) an autoregressive WaveNet [19] *vocoder*, which converts the spectrogram into time domain waveforms.[1]

## 2.1 Speaker encoder

The speaker encoder is used to condition the synthesis network on a reference speech signal from the desired target speaker. Critical to good generalization is the use of a representation which captures the characteristics of different speakers, and the ability to identify these characteristics using only a short adaptation signal, independent of its phonetic content and background noise. These requirements are satisfied using a speaker-discriminative model trained on a text-independent speaker verification task.

We follow [22], which proposed a highly scalable and accurate neural network framework for speaker verification. The network maps a sequence of log-mel spectrogram frames computed from a speech utterance of arbitrary length, to a fixed-dimensional embedding vector, known as *d-vector* [20, 9]. The network is trained to optimize a generalized end-to-end speaker verification loss, so that embeddings of utterances from the same speaker have high cosine similarity, while those of utterances from different speakers are far apart in the embedding space. The training dataset consists of speech audio examples segmented into 1.6 seconds and associated speaker identity labels; no transcripts are used.

Input 40-channel log-mel spectrograms are passed to a network consisting of a stack of 3 LSTM layers of 768 cells, each followed by a projection to 256 dimensions. The final embedding is created by $L_2$-normalizing the output of the top layer at the final frame. During inference, an arbitrary length utterance is broken into 800ms windows, overlapped by 50%. The network is run independently on each window, and the outputs are averaged and normalized to create the final utterance embedding.

Although the network is not optimized directly to learn a representation which captures speaker characteristics relevant to synthesis, we find that training on a speaker discrimination task leads to an embedding which is directly suitable for conditioning the synthesis network on speaker identity.

## 2.2 Synthesizer

We extend the recurrent sequence-to-sequence with attention Tacotron 2 architecture [15] to support multiple speakers following a scheme similar to [8]. An embedding vector for the target speaker is concatenated with the synthesizer encoder output at each time step. In contrast to [8], we find that simply passing embeddings to the attention layer, as in Figure 1, converges across different speakers.

We compare two variants of this model, one which computes the embedding using the speaker encoder, and a baseline which optimizes a fixed embedding for each speaker in the training set, essentially learning a lookup table of speaker embeddings similar to [8, 13].

The synthesizer is trained on pairs of text transcript and target audio. At the input, we map the text to a sequence of phonemes, which leads to faster convergence and improved pronunciation of rare words and proper nouns. The network is trained in a transfer learning configuration, using a pretrained speaker encoder (whose parameters are frozen) to extract a speaker embedding from the target audio, i.e. the speaker reference signal is the same as the target speech during training. No explicit speaker identifier labels are used during training.

Target spectrogram features are computed from 50ms windows computed with a 12.5ms step, passed through an 80-channel mel-scale filterbank followed by log dynamic range compression. We extend [15] by augmenting the $L_2$ loss on the predicted spectrogram with an additional $L_1$ loss. In practice,

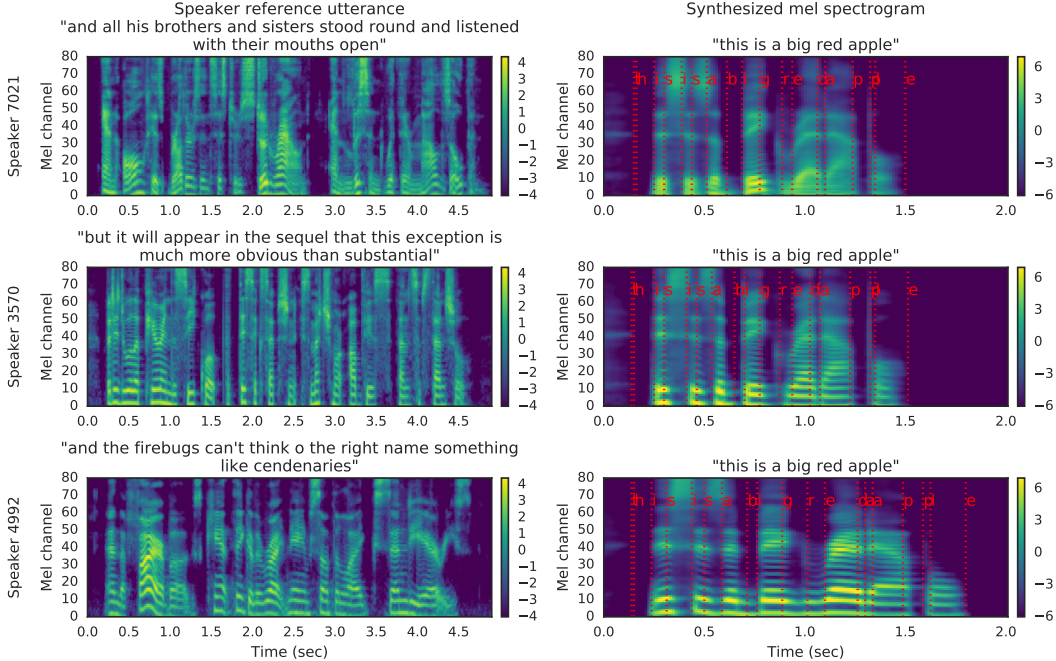

Figure 2: Example synthesis of a sentence in different voices using the proposed system. Mel spectrograms are visualized for reference utterances used to generate speaker embeddings (left), and the corresponding synthesizer outputs (right). The text-to-spectrogram alignment is shown in red. Three speakers held out of the train sets are used: one male (top) and two female (center and bottom).

we found this combined loss to be more robust on noisy training data. In contrast to [10], we don't introduce additional loss terms based on the speaker embedding.

## 2.3 Neural vocoder

We use the sample-by-sample autoregressive WaveNet [19] as a vocoder to invert synthesized mel spectrograms emitted by the synthesis network into time-domain waveforms. The architecture is the same as that described in [15], composed of 30 dilated convolution layers. The network is not directly conditioned on the output of the speaker encoder. The mel spectrogram predicted by the synthesizer network captures all of the relevant detail needed for high quality synthesis of a variety of voices, allowing a multispeaker vocoder to be constructed by simply training on data from many speakers.

## 2.4 Inference and zero-shot speaker adaptation

During inference the model is conditioned using arbitrary untranscribed speech audio, which does not need to match the text to be synthesized. Since the speaker characteristics to use for synthesis are inferred from audio, it can be conditioned on audio from speakers that are outside the training set. In practice we find that using a single audio clip of a few seconds duration is sufficient to synthesize new speech with the corresponding speaker characteristics, representing zero-shot adaptation to novel speakers. In Section 3 we evaluate how well this process generalizes to previously unseen speakers.

An example of the inference process is visualized in Figure 2, which shows spectrograms synthesized using several different 5 second speaker reference utterances. Compared to those of the female (center and bottom) speakers, the synthesized male (top) speaker spectrogram has noticeably lower fundamental frequency, visible in the denser harmonic spacing (horizontal stripes) in low frequencies, as well as formants, visible in the mid-frequency peaks present during vowel sounds such as the 'i' at 0.3 seconds – the top male $F_2$ is in mel channel 35, whereas the $F_2$ of the middle speaker appears closer to channel 40. Similar differences are also visible in sibilant sounds, e.g. the 's' at 0.4 seconds contains more energy in lower frequencies in the male voice than in the female voices. Finally, the characteristic speaking rate is also captured to some extent by the speaker embedding, as can be seen

Table 1: Speech naturalness Mean Opinion Score (MOS) with 95% confidence intervals.

| System | VCTK Seen | VCTK Unseen | LibriSpeech Seen | LibriSpeech Unseen |
|---|---|---|---|---|
| Ground truth | $4.43 \pm 0.05$ | $4.49 \pm 0.05$ | $4.49 \pm 0.05$ | $4.42 \pm 0.07$ |
| Embedding table | $4.12 \pm 0.06$ | N/A | $3.90 \pm 0.06$ | N/A |
| Proposed model | $4.07 \pm 0.06$ | $4.20 \pm 0.06$ | $3.89 \pm 0.06$ | $4.12 \pm 0.05$ |

by the longer signal duration in the bottom row compared to the top two. Similar observations can be made about the corresponding reference utterance spectrograms in the right column.

## 3 Experiments

We used two public datasets for training the speech synthesis and vocoder networks. VCTK [21] contains 44 hours of clean speech from 109 speakers, the majority of which have British accents. We downsampled the audio to 24 kHz, trimmed leading and trailing silence (reducing the median duration from 3.3 seconds to 1.8 seconds), and split into three subsets: train, validation (containing the same speakers as the train set) and test (containing 11 speakers held out from the train and validation sets).

LibriSpeech [12] consists of the union of the two "clean" training sets, comprising 436 hours of speech from 1,172 speakers, sampled at 16 kHz. The majority of speech is US English, however since it is sourced from audio books, the tone and style of speech can differ significantly between utterances from the same speaker. We resegmented the data into shorter utterances by force aligning the audio to the transcript using an ASR model and breaking segments on silence, reducing the median duration from 14 to 5 seconds. As in the original dataset, there is no punctuation in transcripts. The speaker sets are completely disjoint among the train, validation, and test sets.

Many recordings in the LibriSpeech clean corpus contain noticeable environmental and stationary background noise. We preprocessed the target spectrogram using a simple spectral subtraction [4] denoising procedure, where the background noise spectrum of an utterance was estimated as the 10th percentile of the energy in each frequency band across the full signal. This process was only used on the synthesis target; the original noisy speech was passed to the speaker encoder.

We trained separate synthesis and vocoder networks for each of these two corpora. Throughout this section, we used synthesis networks trained on phoneme inputs, in order to control for pronunciation in subjective evaluations. For the VCTK dataset, whose audio is quite clean, we found that the vocoder trained on ground truth mel spectrograms worked well. However for LibriSpeech, which is noisier, we found it necessary to train the vocoder on spectrograms predicted by the synthesizer network. No denoising was performed on the target waveform for vocoder training.

The speaker encoder was trained on a proprietary voice search corpus containing 36M utterances with median duration of 3.9 seconds from 18K English speakers in the United States. This dataset is not transcribed, but contains anonymized speaker identities. It is never used to train synthesis networks.

We primarily rely on crowdsourced Mean Opinion Score (MOS) evaluations based on subjective listening tests. All our MOS evaluations are aligned to the *Absolute Category Rating* scale [14], with rating scores from 1 to 5 in 0.5 point increments. We use this framework to evaluate synthesized speech along two dimensions: its naturalness and similarity to real speech from the target speaker.

### 3.1 Speech naturalness

We compared the naturalness of synthesized speech using synthesizers and vocoders trained on VCTK and LibriSpeech. We constructed an evaluation set of 100 phrases which do not appear in any training sets, and evaluated two sets of speakers for each model: one composed of speakers included in the train set (Seen), and another composed of those that were held out (Unseen). We used 11 seen and unseen speakers for VCTK and 10 seen and unseen speakers for LibriSpeech (Appendix D). For each speaker, we randomly chose one utterance with duration of about 5 seconds to use to compute the speaker embedding (see Appendix C). Each phrase was synthesized for each speaker, for a total of about 1,000 synthesized utterances per evaluation. Each sample was rated by a single rater, and each evaluation was conducted independently: the outputs of different models were not compared directly.

Table 2: Speaker similarity Mean Opinion Score (MOS) with 95% confidence intervals.

| System | Speaker Set | VCTK | LibriSpeech |
|---|---|---|---|
| Ground truth | Same speaker | $4.67 \pm 0.04$ | $4.33 \pm 0.08$ |
| Ground truth | Same gender | $2.25 \pm 0.07$ | $1.83 \pm 0.07$ |
| Ground truth | Different gender | $1.15 \pm 0.04$ | $1.04 \pm 0.03$ |
| Embedding table | Seen | $4.17 \pm 0.06$ | $3.70 \pm 0.08$ |
| Proposed model | Seen | $4.22 \pm 0.06$ | $3.28 \pm 0.08$ |
| Proposed model | Unseen | $3.28 \pm 0.07$ | $3.03 \pm 0.09$ |

Results are shown in Table 1, comparing the proposed model to baseline multispeaker models that utilize a lookup table of speaker embeddings similar to [8, 13], but otherwise have identical architectures to the proposed synthesizer network. The proposed model achieved about 4.0 MOS in all datasets, with the VCTK model obtaining a MOS about 0.2 points higher than the LibriSpeech model when evaluated on seen speakers. This is the consequence of two drawbacks of the LibriSpeech dataset: (i) the lack of punctuation in transcripts, which makes it difficult for the model to learn to pause naturally, and (ii) the higher level of background noise compared to VCTK, some of which the synthesizer has learned to reproduce, despite denoising the training targets as described above.

Most importantly, the audio generated by our model for unseen speakers is deemed to be at least as natural as that generated for seen speakers. Surprisingly, the MOS on unseen speakers is higher than that of seen speakers, by as much as 0.2 points on LibriSpeech. This is a consequence of the randomly selected reference utterance for each speaker, which sometimes contains uneven and non-neutral prosody. In informal listening tests we found that the prosody of the synthesized speech sometimes mimics that of the reference, similar to [16]. This effect is larger on LibriSpeech, which contains more varied prosody. This suggests that additional care must be taken to disentangle speaker identity from prosody within the synthesis network, perhaps by integrating a prosody encoder as in [16, 24], or by training on randomly paired reference and target utterances from the same speaker.

## 3.2 Speaker similarity

To evaluate how well the synthesized speech matches that from the target speaker, we paired each synthesized utterance with a randomly selected ground truth utterance from the same speaker. Each pair is rated by one rater with the following instructions: "You should not judge the content, grammar, or audio quality of the sentences; instead, just focus on the similarity of the speakers to one another."

Results are shown in Table 2. The scores for the VCTK model tend to be higher than those for LibriSpeech, reflecting the cleaner nature of the dataset. This is also evident in the higher ground truth baselines on VCTK. For seen speakers on VCTK, the proposed model performs about as well as the baseline which uses an embedding lookup table for speaker conditioning. However, on LibriSpeech, the proposed model obtained a lower similarity MOS than the baseline, which is likely due to the wider degree of within-speaker variation (Appendix B), and background noise level in the dataset.

On unseen speakers, the proposed model obtains lower similarity between ground truth and synthesized speech. On VCTK, the similarity score of 3.28 is between "moderately similar" and "very similar" on the evaluation scale. Informally, it is clear that the proposed model is able to transfer the broad strokes of the speaker characteristics for unseen speakers, clearly reflecting the correct gender, pitch, and formant ranges (as also visualized in Figure 2). But the significantly reduced similarity scores on unseen speakers suggests that some nuances, e.g. related to characteristic prosody, are lost.

The speaker encoder is trained only on North American accented speech. As a result, accent mismatch constrains our performance on speaker similarity on VCTK since the rater instructions did not specify how to judge accents, so raters may consider a pair to be from different speakers if the accents do not match. Indeed, examination of rater comments shows that our model sometimes produced a different accent than the ground truth, which led to lower scores. However, a few raters commented that the tone and inflection of the voices sounded very similar despite differences in accent.

As an initial evaluation of the ability to generalize to out of domain speakers, we used synthesizers trained on VCTK and LibriSpeech to synthesize speakers from the other dataset. We only varied the train set of the synthesizer and vocoder networks; both models used an identical speaker encoder. As

Table 3: Cross-dataset evaluation on naturalness and speaker similarity for unseen speakers.

| Synthesizer Training Set | Testing Set | Naturalness | Similarity |
|---|---|---|---|
| VCTK | LibriSpeech | $4.28 \pm 0.05$ | $1.82 \pm 0.08$ |
| LibriSpeech | VCTK | $4.01 \pm 0.06$ | $2.77 \pm 0.08$ |

Table 4: Speaker verification EERs of different synthesizers on unseen speakers.

| Synthesizer Training Set | Training Speakers | SV-EER on VCTK | SV-EER on LibriSpeech |
|---|---|---|---|
| Ground truth | – | 1.53% | 0.93% |
| VCTK | 98 | 10.46% | 29.19% |
| LibriSpeech | 1.2K | 6.26% | 5.08% |

shown in Table 3, the models were able to generate speech with the same degree of naturalness as on unseen, but in-domain, speakers shown in Table 1. However, the LibriSpeech model synthesized VCTK speakers with significantly higher speaker similarity than the VCTK model is able to synthesize LibriSpeech speakers. The better generalization of the LibriSpeech model suggests that training the synthesizer on only 100 speakers is insufficient to enable high quality speaker transfer.

## 3.3   Speaker verification

As an objective metric of the degree of speaker similarity between synthesized and ground truth audio for unseen speakers, we evaluated the ability of a limited speaker verification system to distinguish synthetic from real speech. We trained a new *eval-only* speaker encoder with the same network topology as Section 2.1, but using a different training set of 28M utterances from 113K speakers. Using a different model for evaluation ensured that metrics were not only valid on a specific speaker embedding space. We enroll the voices of 21 real speakers: 11 speakers from VCTK, and 10 from LibriSpeech, and score synthesized waveforms against the set of enrolled speakers. All enrollment and verification speakers are unseen during synthesizer training. Speaker verification equal error rates (SV-EERs) are estimated by pairing each test utterance with each enrollment speaker. We synthesized 100 test utterances for each speaker, so 21,000 or 23,100 trials were performed for each evaluation.

As shown in Table 4, as long as the synthesizer was trained on a sufficiently large set of speakers, i.e. on LibriSpeech, the synthesized speech is typically most similar to the ground truth voices. The LibriSpeech synthesizer obtains similar EERs of 5-6% using reference speakers from both datasets, whereas the one trained on VCTK performs much worse, especially on out-of-domain LibriSpeech speakers. These results are consistent with the subjective evaluation in Table 3.

To measure the difficulty of discriminating between real and synthetic speech for the same speaker, we performed an additional evaluation with an expanded set of enrolled speakers including 10 synthetic versions of the 10 real LibriSpeech speakers. On this 20 voice discrimination task we obtain an EER of 2.86%, demonstrating that, while the synthetic speech tends to be close to the target speaker (cosine similarity > 0.6, and as in Table 4), it is nearly always even closer to other synthetic utterances for the same speaker (similarity > 0.7). From this we can conclude that the proposed model can generate speech that resembles the target speaker, but not well enough to be confusable with a real speaker.

## 3.4   Speaker embedding space

Visualizing the speaker embedding space further contextualizes the quantitive results described in Section 3.2 and 3.3. As shown in Figure 3, different speakers are well separated from each other in the speaker embedding space. The PCA visualization (left) shows that synthesized utterances tend to lie very close to real speech from the same speaker in the embedding space. However, synthetic utterances are still easily distinguishable from the real human speech as demonstrated by the t-SNE visualization (right) where utterances from each synthetic speaker form a distinct cluster adjacent to a cluster of real utterances from the corresponding speaker.

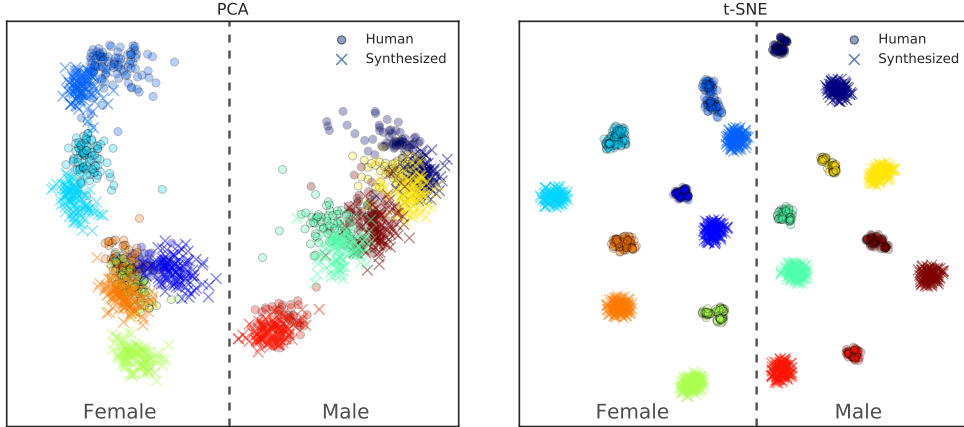

Figure 3: Visualization of speaker embeddings extracted from LibriSpeech utterances. Each color corresponds to a different speaker. Real and synthetic utterances appear nearby when they are from the same speaker, however real and synthetic utterances consistently form distinct clusters.

Table 5: Performance using speaker encoders (SEs) trained on different datasets. Synthesizers are all trained on LibriSpeech Clean and evaluated on held out speakers. LS: LibriSpeech, VC: VoxCeleb.

| SE Training Set | Speakers | Embedding Dim | Naturalness | Similarity | SV-EER |
|---|---|---|---|---|---|
| LS-Clean | 1.2K | 64 | $3.73 \pm 0.06$ | $2.23 \pm 0.08$ | 16.60% |
| LS-Other | 1.2K | 64 | $3.60 \pm 0.06$ | $2.27 \pm 0.09$ | 15.32% |
| LS-Other + VC | 2.4K | 256 | $3.83 \pm 0.06$ | $2.43 \pm 0.09$ | 11.95% |
| LS-Other + VC + VC2 | 8.4K | 256 | $3.82 \pm 0.06$ | $2.54 \pm 0.09$ | 10.14% |
| Internal | 18K | 256 | $4.12 \pm 0.05$ | $3.03 \pm 0.09$ | 5.08% |

Speakers appear to be well separated by gender in both the PCA and t-SNE visualizations, with all female speakers appearing on the left, and all male speakers appearing on the right. This is an indication that the speaker encoder has learned a reasonable representation of speaker space.

## 3.5 Number of speaker encoder training speakers

It is likely that the ability of the proposed model to generalize well across a wide variety of speakers is based on the quality of the representation learned by the speaker encoder. We therefore explored the effect of the speaker encoder training set on synthesis quality. We made use of three additional training sets: (1) LibriSpeech Other, which contains 461 hours of speech from a set of 1,166 speakers disjoint from those in the clean subsets, (2) VoxCeleb [11], and (3) VoxCeleb2 [6] which contain 139K utterances from 1,211 speakers, and 1.09M utterances from 5,994 speakers, respectively.

Table 5 compares the performance of the proposed model as a function of the number of speakers used to train the speaker encoder. This measures the importance of speaker diversity when training the speaker encoder. To avoid overfitting, the speaker encoders trained on small datasets (top two rows) use a smaller network architecture (256-dim LSTM cells with 64-dim projections) and output 64 dimensional speaker embeddings.

We first evaluate the speaker encoder trained on LibriSpeech Clean and Other sets, each of which contain a similar number of speakers. In Clean, the speaker encoder and synthesizer are trained on the same data, a baseline similar to the non-fine tuned speaker encoder from [2], except that it is trained discriminatively as in [10]. This matched condition gives a slightly better naturalness and a similar similarity score. As the number of training speakers increases, both naturalness and similarity improve significantly. The objective EER results also improve alongside the subjective evaluations.

These results have an important implication for multispeaker TTS training. The data requirement for the speaker encoder is much cheaper than full TTS training since no transcripts are necessary, and the audio quality can be lower than for TTS training. We have shown that it is possible to synthesize very

Table 6: Speech from fictitious speakers compared to their nearest neighbors in the train sets. Synthesizer was trained on LS Clean. Speaker Encoder was trained on LS-Other + VC + VC2.

| Nearest neighbors in | Cosine similarity | SV-EER | Naturalness MOS |
|---|---|---|---|
| Synthesizer train set | 0.222 | 56.77% | |
| Speaker Encoder train set | 0.245 | 38.54% | $3.65 \pm 0.06$ |

natural TTS by combining a speaker encoder network trained on large amounts of untranscribed data with a TTS network trained on a smaller set of high quality data.

## 3.6 Fictitious speakers

Bypassing the speaker encoder network and conditioning the synthesizer on random points in the speaker embedding space results in speech from fictitious speakers which are not present in the train or test sets of either the synthesizer or the speaker encoder. This is demonstrated in Table 6, which compares 10 such speakers, generated from uniformly sampled points on the surface of the unit hypersphere, to their nearest neighbors in the training sets of the component networks. SV-EERs are computed using the same setup as Section 3.3 after enrolling voices of the 10 nearest neighbors. Even though these speakers are totally fictitious, the synthesizer and the vocoder are able to generate audio as natural as for seen or unseen real speakers. The low cosine similarity to the nearest neighbor training utterances and very high EER indicate that they are indeed distinct from the training speakers.

## 4 Conclusion

We present a neural network-based system for multispeaker TTS synthesis. The system combines an independently trained speaker encoder network with a sequence-to-sequence TTS synthesis network and neural vocoder based on Tacotron 2. By leveraging the knowledge learned by the discriminative speaker encoder, the synthesizer is able to generate high quality speech not only for speakers seen during training, but also for speakers never seen before. Through evaluations based on a speaker verification system as well as subjective listening tests, we demonstrated that the synthesized speech is reasonably similar to real speech from the target speakers, even on such unseen speakers.

We ran experiments to analyze the impact of the amount of data used to train the different components, and found that, given sufficient speaker diversity in the synthesizer training set, speaker transfer quality could be significantly improved by increasing the amount of speaker encoder training data.

Transfer learning is critical to achieving these results. By separating the training of the speaker encoder and the synthesizer, the system significantly lowers the requirements for multispeaker TTS training data. It requires neither speaker identity labels for the synthesizer training data, nor high quality clean speech or transcripts for the speaker encoder training data. In addition, training the components independently significantly simplifies the training configuration of the synthesizer network compared to [10] since it does not require additional triplet or contrastive losses. However, modeling speaker variation using a low dimensional vector limits the ability to leverage large amounts of reference speech. Improving speaker similarity given more than a few seconds of reference speech requires a model adaptation approach as in [2], and more recently in [5].

Finally, we demonstrate that the model is able to generate realistic speech from fictitious speakers that are dissimilar from the training set, implying that the model has learned to utilize a realistic representation of the space of speaker variation.

The proposed model does not attain human-level naturalness, despite the use of a WaveNet vocoder (along with its very high inference cost), in contrast to the single speaker results from [15]. This is a consequence of the additional difficulty of generating speech for a variety of speakers given significantly less data per speaker, as well as the use of datasets with lower data quality. An additional limitation lies in the model's inability to transfer accents. Given sufficient training data, this could be addressed by conditioning the synthesizer on independent speaker and accent embeddings. Finally, we note that the model is also not able to completely isolate the speaker voice from the prosody of the reference audio, a similar trend to that observed in [16].

**Acknowledgements**

The authors thank Heiga Zen, Yuxuan Wang, Samy Bengio, the Google AI Perception team, and the Google TTS and DeepMind Research teams for their helpful discussions and feedback.

## Footnotes

[1]See https://google.github.io/tacotron/publications/speaker_adaptation for samples.

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
