[Supplementary Material · Transfer_Learning_from_Speaker_Verification_to_Multispeaker_Text_To_Speech_Synthesis - Camera ready - supplement.pdf]

# Appendix A    Additional joint training baselines

Table 7: Speech naturalness and speaker similarity Mean Opinion Score (MOS) with 95% confidence intervals of baseline models where the speaker encoder and synthesizer networks are trained jointly (top two rows). Included for comparison are the separately trained baseline from Table 5 (middle row) as well as the embedding lookup table baseline and proposed model from Tables 1 and 2 (bottom two rows). All but the bottom row, are trained entirely on LibriSpeech. The bottom row uses a speaker encoder trained on a separate speaker corpus. All evaluations are on LibriSpeech.

| System | Embedding Dim | Naturalness MOS | | Similarity MOS | |
|---|---|---|---|---|---|
| | | Seen | Unseen | Seen | Unseen |
| Joint training | 64 | $3.72 \pm 0.06$ | $3.59 \pm 0.07$ | $2.47 \pm 0.08$ | $2.44 \pm 0.09$ |
| Joint training + speaker loss | 64 | $3.71 \pm 0.06$ | $3.71 \pm 0.06$ | $2.82 \pm 0.08$ | $2.12 \pm 0.08$ |
| Separate training (Table 5) | 64 | $3.88 \pm 0.06$ | $3.73 \pm 0.06$ | $2.64 \pm 0.08$ | $2.23 \pm 0.08$ |
| Embedding table (Tables 1,2) | 64 | $3.90 \pm 0.06$ | N/A | $3.70 \pm 0.08$ | N/A |
| Proposed model (Tables 1,2,5) | 256 | $3.89 \pm 0.06$ | $4.12 \pm 0.05$ | $3.28 \pm 0.08$ | $3.03 \pm 0.09$ |

Although separate training of the speaker encoder and synthesizer networks is necessary if the speaker encoder is trained on a larger corpus of untranscribed speech, as described in Section 3.5, in this section we evaluate the effectiveness of joint training of the speaker encoder and synthesizer networks as a baseline, similar to [10].

We train on the Clean subset of LibriSpeech, containing 1.2K speakers, and use a speaker embedding dimension of 64 following Section 3.5. We compare two baseline jointly-trained systems: one without any constraints on the output of the speaker encoder, analogous to [16], and another with an additional speaker discrimination loss formed by passing the 64 dimension speaker embedding through a linear projection to form the logits for a softmax speaker classifier, optimizing a corresponding cross-entropy loss.

Naturalness and speaker similarity MOS results are shown in Table 7, comparing these jointly trained baselines to results reported in previous sections. We find that both jointly trained models obtain similar naturalness MOS on Seen speakers, with the variant incorporating a discriminative speaker loss performing better on Unseen speakers. In terms of both naturalness and similarity on Unseen speakers, the model which includes the speaker loss has nearly the same performance as the baseline from Table 5, which uses a separately trained speaker encoder that is also optimized to discriminate between speakers. Finally, we note that the proposed model, which uses a speaker encoder trained separately on a corpus of 18K speakers, significantly outperforms all baselines, once again highlighting the effectiveness of transfer learning for this task.

# Appendix B    Speaker variation

The tone and style of LibriSpeech utterances varies significantly between utterances even from the same speaker. In some examples, the speaker even tries to mimic a voice in a different gender. As a result, comparing the speaker similarity between different utterances from a same speaker (i.e. self-similarity) can sometimes be relatively low, and varies significantly speaker by speaker. Because of the noise level in LibriSpeech recordings, some speakers have significantly lower naturalness scores. This again varies significantly speaker by speaker. This can be seen in Table 8. In contrast, VCTK is more consistent in terms of both naturalness and self-similarity.

Table 4 shows the variance in naturalness MOS across different speakers on synthesized audio. It compares the MOS of different speakers for both ground truth and synthesized on VCTK, revealing that the performance of our proposed model on VCTK is also very speaker dependant. For example, speaker "p240" obtained a MOS of 4.48, which is very close to the MOS of the ground truth (4.57), but speaker "p260" is a full 0.5 points behind its ground truth.

Table 8: Ground truth MOS evaluations breakdown on unseen speakers. Similarity evaluations compare two utterances by the same speaker.

(a) VCTK

| Speaker | Gender | Naturalness | Similarity |
|---------|--------|-------------|------------|
| p230 | F | 4.22 | 4.65 |
| p240 | F | 4.57 | 4.67 |
| p250 | F | 4.31 | 4.72 |
| p260 | M | 4.56 | 4.31 |
| p270 | M | 4.29 | 4.77 |
| p280 | F | 4.41 | 4.71 |
| p300 | F | 4.60 | 4.87 |
| p310 | F | 4.56 | 4.52 |
| p330 | F | 4.34 | 4.77 |
| p340 | F | 4.44 | 4.71 |
| p360 | M | 4.36 | 4.63 |

(b) LibriSpeech

| Speaker | Gender | Naturalness | Similarity |
|---------|--------|-------------|------------|
| 1320 | M | 4.64 | 4.43 |
| 2300 | M | 4.67 | 4.22 |
| 3570 | F | 4.31 | 4.38 |
| 3575 | F | 4.59 | 4.36 |
| 4970 | F | 3.77 | 4.16 |
| 4992 | F | 4.40 | 3.81 |
| 6829 | F | 4.24 | 4.39 |
| 7021 | M | 4.71 | 4.55 |
| 7729 | M | 4.55 | 4.48 |
| 8230 | M | 4.65 | 4.70 |

Figure 4: Per-speaker naturalness MOS of ground truth and synthesized speech on unseen VCTK speakers.

## Appendix C   Impact of reference speech duration

Table 9: Impact of duration of reference speech utterance. Evaluated on VCTK.

|  | 1 sec | 2 sec | 3 sec | 5 sec | 10 sec |
|--|-------|-------|-------|-------|--------|
| Naturalness (MOS) | $4.28 \pm 0.05$ | $4.26 \pm 0.05$ | $4.18 \pm 0.06$ | $4.20 \pm 0.06$ | $4.16 \pm 0.06$ |
| Similarity (MOS) | $2.85 \pm 0.07$ | $3.17 \pm 0.07$ | $3.31 \pm 0.07$ | $3.28 \pm 0.07$ | $3.18 \pm 0.07$ |
| SV-EER | 17.28% | 11.30% | 10.80% | 10.46% | 11.50% |

The proposed model depends on a reference speech signal fed into the speaker encoder. As shown in Table 9, increasing the length of the reference speech significantly improved the similarity, because we can compute more precise speaker embedding with it. Quality saturates at about 5 seconds on VCTK. Shorter reference utterances give slightly better naturalness, because they better match the durations of reference utterances used to train the synthesizer, whose median duration is 1.8 seconds. The proposed model achieves close to the best performance using only 2 seconds of reference audio. The performance saturation using only 5 seconds of speech highlights a limitation of the proposed model, which is constrained by the small capacity of the speaker embedding. Similar scaling was found in [2], where adapting a speaker embedding alone was shown to be effective given limited adaptation data, however fine tuning the full model was required to improve performance if more data was available. This pattern was also confirmed in more recent work [5].

# Appendix D  Evaluation speaker sets

Table 10: Speaker sets used for evaluation.

(a) VCTK

| | | | | | Seen | | | | | | |
|---|---|---|---|---|---|---|---|---|---|---|---|
| Speaker | p231 | p241 | p251 | p261 | p271 | p281 | p301 | p311 | p341 | p351 | p361 |
| Gender | F | M | M | F | M | M | F | M | F | F | F |
| | | | | | Unseen | | | | | | |
| Speaker | p230 | p240 | p250 | p260 | p270 | p280 | p300 | p310 | p330 | p340 | p360 |
| Gender | F | F | F | M | M | F | F | F | F | F | M |

(b) LibriSpeech

| | | | | | Seen | | | | | |
|---|---|---|---|---|---|---|---|---|---|---|
| Speaker | 446 | 1246 | 2136 | 4813 | 4830 | 6836 | 7517 | 7800 | 8238 | 8123 |
| Gender | M | F | M | M | M | M | F | F | F | F |
| | | | | | Unseen | | | | | |
| Speaker | 1320 | 2300 | 3570 | 3575 | 4970 | 4992 | 6829 | 7021 | 7729 | 8230 |
| Gender | M | M | F | F | F | F | F | M | M | M |

# Appendix E  Fictitious speakers

Figure 5: Example synthesis of a sentence conditioned on several random speaker embeddings sampled from the unit hypersphere. All samples contain consistent phonetic content, but there is clear variation in fundamental frequency and speaking rate. Audio files corresponding to these utterances are included in the demo page (`https://google.github.io/tacotron/publications/speaker_adaptation`).

## Appendix F    Speaker similarity MOS evaluation interface

# Instructions

In this task, your job is to evaluate if the two speech audio samples are from the same speaker. Please release this task if any of the following are true:

- You think you do not have good listening ability.
- There is considerable background noise (street noise, loud fan/air-conditioner, open TV/radio, people talking, etc).
- For any reason, you can't hear the audio samples.

# Task

**How are you listening to the speech samples?**

🔘 **Headphones, with no noise in the background.** I am listening to the speech sample using headphones and there is **no noise** around me (people talking, music playing, air-conditioners, fans, etc.).

⚪ **Headphones, with some low-level noise in the background.** I am listening to the speech sample using headphones and there is some **low-level** noise around me (people talking, music playing, air-conditioners, fans, etc.).

⚪ **Audio speakers or other.**

---

**Speaker Similarity Between Two Speech Samples**

Please listen to the two speech samples below (Sample A and Sample B) and rate how similar they are. Your rating should reflect your evaluation of how close the voices of the two speakers sound. You **should not judge the content, grammar, or audio quality** of the sentences; instead, just focus on the similarity of the speakers to one another.

Speech samples (please listen at least **two times each**)

| Sample A | ▶  0:00 / 0:02  ●———  🔊  —● |
| --- | --- |
| Sample B | ▶  0:00 / 0:03  ●———  🔊  —● |

Please rate the similarity of the two speech samples

├———┼———┼———┼———┼———┼━━━━┃┼———┼———┤

N/A          Not at all similar     Slightly similar     Moderately similar     Very similar     Extremely similar

Comment (optional)

Figure 6: Interface of MOS evaluation for speaker similarity.