[Reviews · NeurIPS 2018]

Reviewer 1



This work offers a clearly defined extension to TTS systems allowing to build good quality voices (even unseen ones during training of either component) from a few adaptation data-points. Authors do not seem to offer any truly new theoretical extension to "building blocks" of their system, which is based on known components proposed elsewhere (speaker encoder, synthesizer and vocoder are based on previously published models). However, their mutual combination is clever, well-engineered and allows building blocks to by independently estimated in either unsupervised (speaker encoder, where audio transcripts are not needed) or supervised (speech synthesizer) ways, on different corpora. This allows for greater flexibility, reducing at the same time requirements for large amounts of transcribed data for each of the components (i.e. speaker embedding nnet can be more robustly trained on larger amounts of real data to produce good quality speaker embeddings which are later used to condition speech synthesizer for target TTS log-mel spectrograms for the given target voice). Good points: - clear, fair and convincing experiments - trained and evaluated on public corpora, which greatly increases reproducibility (portion of the experiments is carried on proprietary data, but all have equivalent experiments constrained to publicly available data) Weak points: - it would probably make sense to investigate the additional adaptability in case one gets more data per speaker, it seems your system cannot easily leverage more than 10s of reference speech data Summary: this is a very good study on generating multi-speaker TTS systems from small amounts of target speaker data. Authors also decouple different models from each other, allowing for greater flexibility on how to estimate each of the working blocks. My score is somewhere between 8 and 7, I gave 8 because this one is clearly the best paper in my batch, but I understand such specific application study may be of limited interest to the community as a whole. At the same time, it's not my role to decide and leave it to area chair(s). ===================================================== Post-rebuttal update: thanks for addressing my suggestions. I keep my score.

Reviewer 2



This paper discusses a voice cloning TTS system that is based on speaker embeddings as opposed to speaker adaptation. The speaker embeddings are found using a speaker encoder. The key idea of this paper is that the speaker encoder is trained separately on a speaker verification task instead of training it together with the TTS system. This reduces the amount of speech-transcript pairs required to train the complete system. The performance of the proposed system is extensively analysed and compared to the performance of a similar system using fixed embeddings for seen speakers, based on the speaker ID. It is shown that the proposed system achieves similar performance for seen speakers, but is also able to generalise to unseen speakers, which is not possible for the fixed embedding system. The quality of the paper is pretty good, but is lacking key baseline systems. The experiments section is extensive. Both quantitative and qualitative results are presented and several evaluation schemes are used. What I feel is lacking are additional baseline systems. The paper should definitely include a baseline where the speaker encoder is not trained on a speaker verification task, but is trained jointly, like in [1] or [8]. Otherwise we cannot assess the value of training the speaker encoder separately. If such a baseline is added we can also compare the results for unseen speakers. Adding a speaker adaptation system as a baseline could also strengthen the paper, but is not as crucial since a comparison has already been done in [1]. The author has made some good points in their response about the experimentation, so I would say that the argumentation is sufficient. The clarity of the paper is good. The core concepts of the paper are explained well and references are extensively cited as comparison with related work. The components of the system are not explained in full detail but references are provided to papers that do explain the details. If the components and hyper parameters of the proposed model are the same as in the reference material the system should be reproducible. I would say the originality and significance of the paper are mediocre. There is not a lot of novelty in the paper. Pre-training the speaker encoder has been done in [5], but a different type of speaker encoder is used. All the other components in the system are not novel. I think putting the components together as proposed in the paper makes sense, so in my opinion it is an interesting system for this task. I would say that it is likely that this system will be used for voice cloning TTS systems in the future.

Reviewer 3



This paper proposes “voice cloning” system to synthesize the voice from unseen speaker during training. It has three separately trained components: (1) a speaker encoder trained on speaker verification task using an large-scale untranscribed dataset; (2) a multi-speaker Tacotron 2, which is a text-to-spectrogram model conditioned on the speaker embedding from speaker encoder; (3) a WaveNet vocoder. Detailed comment: 1, In speaker encoder, what is the similarity metric used for speaker verification training? 2, Any particular reason to use 40-channel Mel in speaker encoder instead of 80-channel as in synthesizer? 3, The training dataset consists of 1.6 seconds audio clip, while an arbitrary length utterance is broken into 800ms windows, overlapped by 50% at inference. It seems to introduce additional training test mismatch. Why not use the same setting? e.g., also apply overlapped 800ms windows in training? 4, During synthesizer training, speaker encoder doesn't need to be constrained by only using target audio as input. Have you tried to feed different utterances from the same speaker into speaker encoder to extract speaker embedding? 5, In Table 1, higher MOS on unseen speakers seems not an overfitting problem, considering the definition of overfitting. It'll be good to check different held-out speaker sets to reduce MOS variance; randomly sampled “easy” speakers might be the issue. Overall, I think this is a good work given the impressive results. The novelty might be limited, as it does not propose a brand new architecture or method. However, it made solid contribution to neural TTS.